# *Staphylococcus aureus* binding to Seraph® 100 Microbind® Affinity Filter: Effects of surface protein expression and treatment duration

Malin-Theres Seffer[1,2], Martin Weinert[1,3], Gabriella Molinari[4], Manfred Rohde[4], Lothar Gröbe[5], Jan T. Kielstein[2‡], Susanne Engelmann[1,3‡]*

1 Helmholtz Centre for Infection Research, Microbial Proteomics, Braunschweig, Germany, 2 Medical Clinic V, Nephrology, Rheumatology, Blood Purification, Academic Teaching Hospital Braunschweig, Braunschweig, Germany, 3 Technische Universität Braunschweig, Institute for Microbiology, Braunschweig, Germany, 4 Helmholtz Centre for Infection Research, Central Facility of Microscopy, Braunschweig Germany, 5 Helmholtz Centre for Infection Research, Experimental Immunology, Braunschweig, Germany

‡ JTK and SE authors contributed equally and should be considered last authors.
* Susanne.Engelmann@helmholtz-hzi.de

**Data Availability Statement:** All relevant data are within the paper and its Supporting Information files.

## Abstract

### Introduction

Extracorporeal blood purification systems represent a promising alternative for treatment of blood stream infections with multiresistant bacteria.

### Objectives

The aim of this study was to analyse the binding activity of *S. aureus* to Seraph affinity filters based on heparin coated beads and to identify effectors influencing this binding activity.

### Results

To test the binding activity, we used *gfp*-expressing *S. aureus* Newman strains inoculated either in 0.9% NaCl or in blood plasma and determined the number of unbound bacteria by FACS analyses after passing through Seraph affinity filters. The binding activity of *S. aureus* was clearly impaired in human plasma: while a percent removal of 42% was observed in 0.9% NaCl (p-value 0.0472) using Seraph mini columns, a percent removal of only 10% was achieved in human plasma (p-value 0.0934). The different composition of surface proteins in *S. aureus* caused by the loss of SarA, SigB, Lgt, and SaeS had no significant influence on its binding activity. In a clinically relevant approach using the Seraph® 100 Microbind® Affinity Filter and 1000 ml of human blood plasma from four different donors, the duration of treatment was shown to have a critical effect on the rate of bacterial reduction. Within the first four hours, the number of bacteria decreased continuously and the reduction in bacteria reached statistical significance after two hours of treatment (percentage reduction 64%, p-value 0.01165). The final reduction after four hours of treatment was close to 90% and is dependent on donor. The capacity of Seraph® 100 for *S. aureus* in human plasma was approximately 5 x 10$^8$ cells.

**Funding:** The author(s) received no specific funding for this work.

**Competing interests:** The authors have declared that no competing interests exist.

## Conclusions

The Seraph affinity filter, based on heparin-coated beads, is a highly efficient method for reducing *S. aureus* in human blood plasma, with efficiency dependent on blood plasma composition and treatment duration.

## Introduction

Sepsis is a serious worldwide health threat that has been reported to affect annually 30 to 50 million people worldwide. Despite international guidelines for treatment of sepsis, the mortality rate for severe sepsis and septic shock remains high with 43.6% and 58.8%, respectively [1,2]. Therefore, the WHO classifies sepsis as one of the major global health problems [3]. Bacterial infections play an important role as triggers for an inadequate response of the immune system, which may lead to life-threatening organ dysfunctions. Due to the steady increase in resistant bacteria, the treatment of bacterial infections has become increasingly difficult in recent years.

*Staphylococcus aureus* is one of the most common and dangerous pathogens, causing a range of infections from mild skin and soft tissue infections to serious systemic infections such as pneumonia, endocarditis or sepsis [4,5]. Patients with surgical wounds and catheters used for outpatient peritoneal dialysis and hemodialysis have a higher risk of becoming infected with this pathogen. Existing *S. aureus* colonisation of the nasopharynx is the main source of subsequent infections [6,7]. Very frequently, these infections are associated with multi-resistant *S. aureus* isolates that are extremely difficult to treat. To underline the seriousness of the situation, in 2017, a WHO expert panel identified *S. aureus* as a pathogen with high priority for research, discovery, and development of new antibiotics [8].

Antibiotics, once our most effective weapon against bacterial infections, are becoming less effective and the introduction of new antibiotics is a long and very expensive process. Therefore, there is an urgent need for alternative treatment options. A highly attractive alternative is the physical reduction of bacteria using an extracorporeal device that purifies circulating body fluids [9–13]. In 2019, the first biomimetic haemoperfusion device able to reduce the number of pathogens in the blood, the Seraph® 100 Microbind® Affinity Filter (ExThera Medical, Martinez, CA, USA), was licensed by the European Union. It uses covalently bound heparin-coated ultra-high molecular weight polyethylene to remove bacteria and viruses from the blood by exploiting their ability to interact with heparin [13]. Heparin is a well-known anticoagulant drug. It mimics heparan sulfate, which is normally expressed on the surface of the cells [14,15]. Bacteria, viruses, and toxins adhere to human cells via heparan sulfate, mainly by charge or electrostatic interaction [14,16,17]. Binding to heparin-coated beads has been demonstrated not only for bacteria known to express heparin-binding proteins such as *S. aureus* and *Staphylococcus epidermidis*, but also for Gram-negative bacteria such as *Klebsiella pneumoniae* or *Escherichia coli* [13,18,19]. Recent studies suggest that also viruses or specific viral components, such as the spike protein and the N-protein of SARS-CoV-2, can also be removed from blood by binding to the Seraph® 100 Microbind® Affinity Filter [20–22].

The aim of this study was first to investigate the effect of human blood plasma on the binding activity of *S. aureus* to the heparin-coated beads of Seraph micro columns compared to 0.9% NaCl. Another aim was to test whether altered expression of proteins on the surface of *S. aureus* would lead to altered binding properties of the bacterium to the heparin-coated beads of Seraph micro columns. And finally, the binding properties of *S. aureus* in human plasma

were analysed under clinically relevant conditions using the Seraph® 100 Microbind® Affinity Filter connected to a hemofiltration device and tubing system.

## Material and methods

### Bacterial strains, plasmids and human blood plasma

*S. aureus* strains and plasmids used in this study are listed in **Table 1**. The *sarA* deletion mutant of strain Newman (Newman Δ*sarA*) was constructed by transduction of the *sarA*:: Tn9J7LTV1 mutation from strain R [23] to strain Newman wild-type [24,25] using phage Φ80 [26]. To generate *gfp*-expressing *S. aureus* strains, the *gfp*-containing plasmid pCtufgfp [27] was transferred into *S. aureus* Newman wild-type [24,25] and its isogenic mutants by phage transduction with bacteriophage Φ80 [26]. The medium was supplemented with chloramphenicol (10 μg/mL) for selection of plasmid containing strains.

Plasma samples were obtained from patients after a routine therapeutic plasma exchange and stored up to 48 hours at 4°C. Written informed consent was obtained from the volunteers who donated their plasma. The study protocol was approved by the Ethics Committee of the Hannover Medical School and was conducted in accordance with the Declaration of Helsinki and German federal guidelines. For micro column experiments, 10 mL aliquots of human blood plasma from one of the donors were stored at -20°C for at least 10 days.

### *S. aureus* binding experiments with miniaturized Seraph adsorbers (micro columns)

*S. aureus* strains were cultivated in chemically defined medium (CDM) [31,32] at 37°C and 120 revolutions per minute (rpm) to an optical density at 500 nm ($OD_{500}$) of 0.5. Miniaturized Seraph adsorbers (micro columns) with a volume of 2.5 mL or 5 mL, which are hand-packed with media and sterilized by ethylenoxide, were provided by ExThera Medical (Martinez, CA, USA). Micro columns with a volume of 2.5 mL were used to study the binding activity of different *S. aureus* mutants in 0.9% NaCl, and micro columns with a volume of 5 mL were used to evaluate the influence of human blood plasma compared to 0.9% NaCl on the bacterial binding activity. The micro columns were first pretreated with 5 mL (2.5 mL micro columns) or 10 mL (5 mL micro columns) 0.9% NaCl solution, then loaded with 2.5 mL (2.5 mL micro columns) or 5 mL (5 mL micro columns) 0.9% NaCl or 5 mL (5 mL micro columns) human plasma containing approx. $3 \times 10^5$ bacteria/mL. Human plasma from one donor was used for these experiments. Subsequently, micro columns were washed twice with 2.5 mL (2.5 mL

**Table 1. Bacterial strains used in this study.**

| Strains | Genotype or characteristics | Reference |
|---|---|---|
| *S. aureus* Newman | wild-type | [24,25] |
| *S. aureus* R | RN6390 *sar*::*Tn9J7LTV1* | [23] |
| *S. aureus* Newman Δ*lgt* | *lgt*::*ermB*, isogenic to strain Newman | [28] |
| *S. aureus* Newman Δ*sarA* | *sarA*::Tn9J7LTV1, isogenic to strain Newman | This study |
| *S. aureus* Newman Δ*sigB* | *sigB*::*ermB*, isogenic to strain Newman | [29] |
| *S. aureus* Newman Δ*saeS* | *saeS*::Tn917, isogenic to strain Newman | [30] |
| *S. aureus* Newman *gfp*⁺ | *S. aureus* Newman containing pCtufgfp | This study |
| *S. aureus* Newman Δ*lgt gfp*⁺ | *S. aureus* Newman Δ*lgt* containing pCtufgfp | This study |
| *S. aureus* Newman Δ*sarA gfp*⁺ | *S. aureus* Newman Δ*sarA* containing pCtufgfp | This study |
| *S. aureus* Newman Δ*sigB gfp*⁺ | *S. aureus* Newman Δ*sigB* containing pCtufgfp | This study |
| *S. aureus* Newman Δ*saeS gfp*⁺ | *S. aureus* Newman Δ*saeS* containing pCtufgfp | This study |

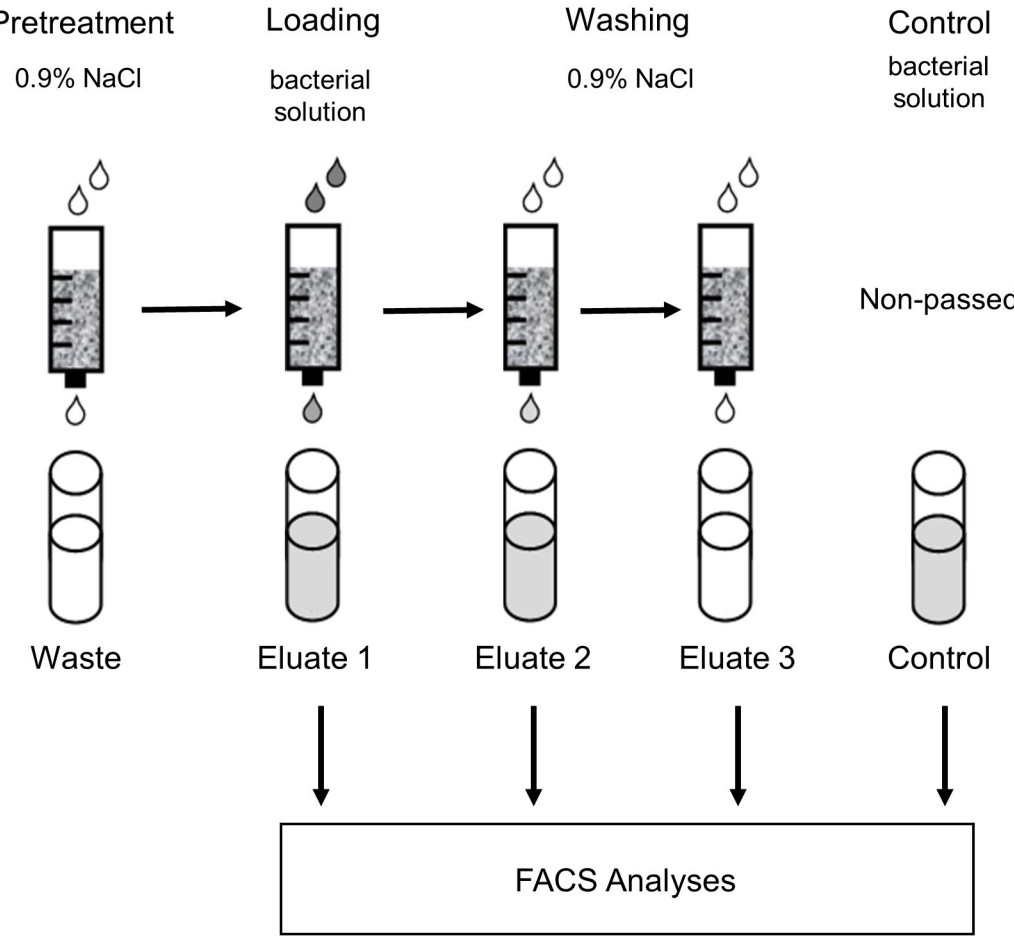

**Fig 1. Schematic overview of the experimental setup using micro columns.** After pretreatment with 0.9% NaCl solution, micro columns were loaded with bacterial solutions containing approx. $3 \times 10^5$ bacteria/ml and washed two times with 0.9% NaCl solution. Eluate 1, 2 and 3 and the control samples were analysed by FACS.

micro columns) or 5 mL (5 mL micro columns) 0.9% NaCl to completely remove unbound bacteria. Free flows of the loading step (Eluate 1) and of each washing step (Eluate 2 and 3) were collected and analysed by FACS. Non-passed bacterial suspensions either in 0.9% NaCl or in blood plasma served as control. Each experiment was performed in triplicate. A schematic overview of the experimental workflow is presented in **Fig 1**.

## *S. aureus* binding experiments with Seraph® 100 Microbind® Affinity blood adsorber

*S. aureus* Newman was cultivated as described above. The Seraph® 100 Microbind® Affinity blood adsorber (ExThera Medical, Martinez, CA, USA), which is also used in clinical applications, was attached to an AFERsmart hemofiltration device (Medica S.a.P., Italia) and a tubing system (Meise Medizintechnik GmbH, Schalksmühle, Germany) thereby establishing a closed circuit of human plasma with constant flow rates as described by Schmidt and coworkers [33] with the following modifications: A flow rate of 60 mL/min was applied and the Seraph® 100 Microbind® Affinity blood adsorbers were pretreated with 500 mL 0.9% NaCl solution. After pretreatment, bags containing 1000 ml human blood plasma were initially inoculated with

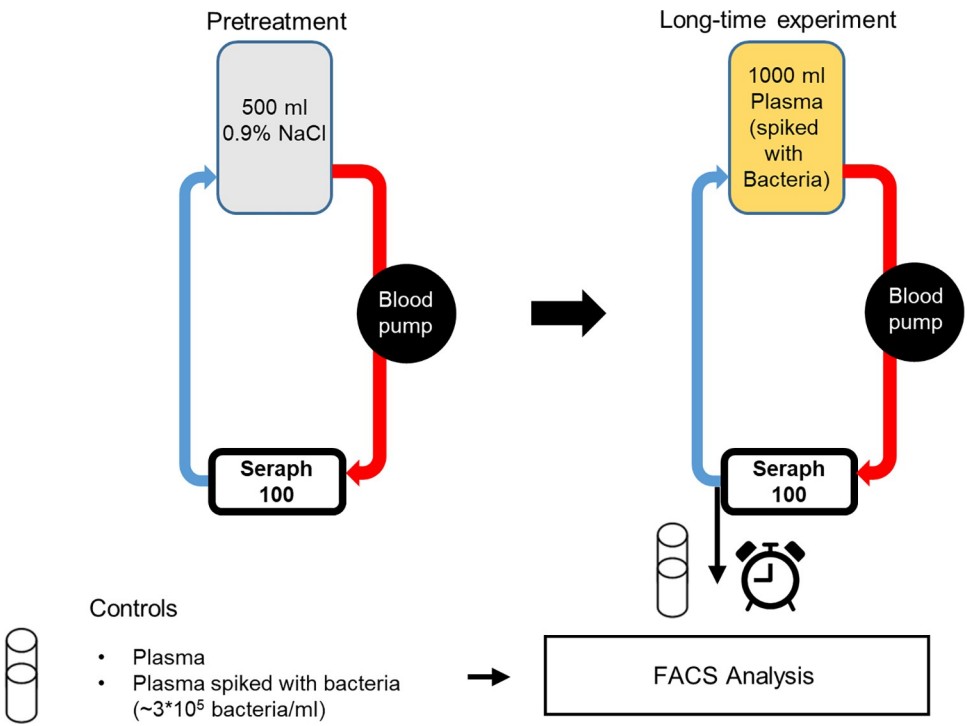

**Fig 2. Schematic overview of the experimental setup using the Seraph® 100 Microbind® Affinity blood adsorber.**
After pretreatment of the Seraph® 100 Microbind® Affinity blood adsorber with 500 ml 0.9% NaCl solution, 1000 ml
plasma were spiked with approx. $4.46 \times 10^5$ bacteria/ml and circulated through the Seraph cartridge by a roller pump.
Plasma samples were obtained at different time points (0, 10, 60, 120, 240, 360, 480 and 600 min) and analysed by
FACS.

approx. $6.1 \times 10^8$ bacteria. Samples were taken before (0 min) and at defined time points (10
min, 60 min, 120 min, 240 min, 360 min, 480 min, 600 min) after starting circulation and
stored on ice until FACS analysis. A schematic overview of the experimental workflow is pro-
vided in **Fig 2**. The experiment was performed four times, each time with a different plasma
bag from a different donor.

## Bacterial cell counting and enumeration assay

To determine numbers of *gfp*-expressing bacteria in a defined sample volume, fluorescence
activated cell sorting (FACS) analyses were performed using an LSR II flow cytometer (Bec-
ton-Dickinson™) and calibrated flow cytometry cell counting bead solution (6 μm diameter)
according to the manufacturer's instructions (ThermoFisher Scientific, Waltham, MA, USA,
Invitrogen). In brief, the samples were homogenized by shaking and diluted in 0.9% NaCl (1:2
for NaCl samples and 1:5 for plasma samples). The counting bead solution was homogenized
by shaking and treatment in an ultrasonic bath (5 min at room temperature, 80 Hz). 5 μL ali-
quotes of the counting bead solution were added to 1000 μL of each cell sample. Immediately
before FACS analyses, samples were shaken for >5 s.

The cytometer was set up for detecting forward scatter (FSC; bandpass filter 488+/-5 nm),
side scatter (SSC; bandpass filter 488+/-5 nm) and green fluorescence (excitation 488 nm,
bandpass filter 530+/-15 nm), all in logarithmic scale. SSC served as the trigger signal. Beads
were gated by forward scatter (FSC) and side scatter (SSC). The *gfp*-expressing bacteria were
gated by green fluorescence and SSC. 10.000 bead events were recorded and served as

reference for sample volume. The respective number of detected bacteria was used to calculate the bacterial cell concentration in the samples.

## Data analyses and statistics

The bacterial cell concentration based on detected GFP-signals was calculated by using the following formula:

$$C = \frac{N_{GFP} * f_{elu}}{N_{beads} * V_b}$$

[$N_{GFP}$ = Number of detected GFP-signals; $f_{elu}$ = eluting factor; $N_{beads}$ = number of detected bead events; $V_b$ = liquid volume].

In the case of experiments with micro columns, the sum of the bacterial counts of the three eluates represents the total number of unbound bacteria (**Fig 1**). The bacterial count of the original bacterial solution was set to 100% to calculate the percent removal.

Statistical analyses of the data were performed with Perseus (version 2.0.3.0, https://maxquant.net/perseus) using p-value based ANOVA and pairwise Student's t-test. The p-value threshold was set at 0.05. All values were first converted to base 2 logarithimic numbers to ensure a normal distribution. Values were then standardised using a z-score to equalise variance. ANOVA was used to test whether the cell numbers changed significantly between more than two strains or sampling points. Significantly altered profiles were then used for paired t-test analysis to compare the standardised values of the samples at each time point with those of the control sample at $t_0$ or of the different strains. The paired t-test was used directly to compare bacterial cell numbers before and after the passage through Seraph micro columns.

## Field Emission Scanning Electron Microscopy (FESEM)

*S. aureus* was cultivated in synthetic medium to an optical density of 2.4. 500 μL culture volume (~ 1 x $10^8$ bacteria) was added to 25 mg heparin-coated beads and fixed with 2% glutaraldehyde in medium. The beads were placed on filter paper in a funnel and washed with TE buffer (20 mM TRIS, 1 mM EDTA, pH 6.9). Dehydration was achieved with a graded series of ethanol washes followed by a gradient of tetramethylsilan (TMS) by gently adding the different liquids dropwise through the filter paper. After air-drying, samples were mounted on aluminium stubs with adhesive tape and covered with a gold/palladium film by sputter coating (SCD 500 Bal-Tec, Liechtenstein) before examination in a field emission scanning electron microscope Zeiss Merlin (Zeiss, Oberkochen). Images were taken using SEM software version 5.05 at an acceleration voltage of 5 kV with the Inlens SE-detector and the HESE2 SE-detector in a 75:25 ratio.

## Results

### Binding activity of *S. aureus* Newman to Seraph micro columns

To determine the binding activity of *S. aureus* to the heparin-coated beads, Seraph micro columns were loaded with a defined number of bacterial cells. For these experiments, the *gfp*-expressing *S. aureus* Newman was used as a reference strain and suspended either in 0.9% NaCl solution or human blood plasma. In total, 1.13 x $10^6$ *S. aureus* cells suspended in NaCl were bound to Seraph micro columns (5 mL), corresponding to 42% of the original bacterial solution (= 2.7 x $10^6$ cells) (**Table 2**). T-test analysis showed that Seraph micro column treatment had a statistically significant effect on *S. aureus* cell counts in NaCl (p-value = 0.0472) (**S1 Table**). *S. aureus* binding to heparin-coated beads was visualized by electron microscopy

**Table 2. Binding activity of *S. aureus* Newman to Seraph micro columns.**

|  | original bacterial solution | bound bacteria |  |  |
|---|---|---|---|---|
|  | (bacterial cell number ± SEM) | (bacterial cell number ± SEM) | t-test (p-value) | percent removal |
| in 0.9% NaCl | $2.70 \times 10^6 \pm 4.3 \times 10^5$ | $1.13 \times 10^6 \pm 4.3 \times 10^5$ | + (0.0472) | 42% |
| in human plasma | $2.85 \times 10^6 \pm 2.5 \times 10^4$ | $2.77 \times 10^5 \pm 1.43 \times 10^5$ | - (0.0934) | 10% |

(**Fig 3**). When using *S. aureus* Newman suspended in human plasma, the binding activity of *S. aureus* to the Seraph micro columns was clearly reduced. Under these conditions, only 10% (= $2.77 \times 10^5$) of the *S. aureus* cells from the original solution (= $2.85 \times 10^6$ bacterial cells) were bound to the beads. According to t-test the analysis, the decrease in the number of *S. aureus* cells in this case was not statistically significant (p-value = 0.0934) (**Tables 2 and S1**).

## Impact of surface protein expression in *S. aureus* on its binding activity to Seraph micro columns

The composition of the surface proteome is highly variable among different *S. aureus* isolates and might be relevant for its binding activity to Seraph [34–37]. To prove this, the binding activity of *S. aureus* Newman wild-type was compared to that of isogenic mutants with strong effects on surface protein expression. Among them are mutants in *sarA*, *saeS*, *lgt* and *sigB*. While the loss of the prolipoprotein diacylglyceryl transferase Lgt completely prevents the attachment of lipoproteins to the cell surface in *S. aureus* [28], the lack of regulatory proteins such as SarA, SigB and SaeS affects the expression of specific sets of genes encoding surface associate proteins in this pathogen [35,36,38–42]. For these experiments, miniaturized Seraph adsorbers (2.5 mL) were loaded with *S. aureus* wild-type or mutant strains suspended in 0.9% NaCl. Percent removal for the Δ*lgt*, Δ*sigB*, Δ*sarA*, and Δ*saeS* mutants were 54%, 55%, 70% and

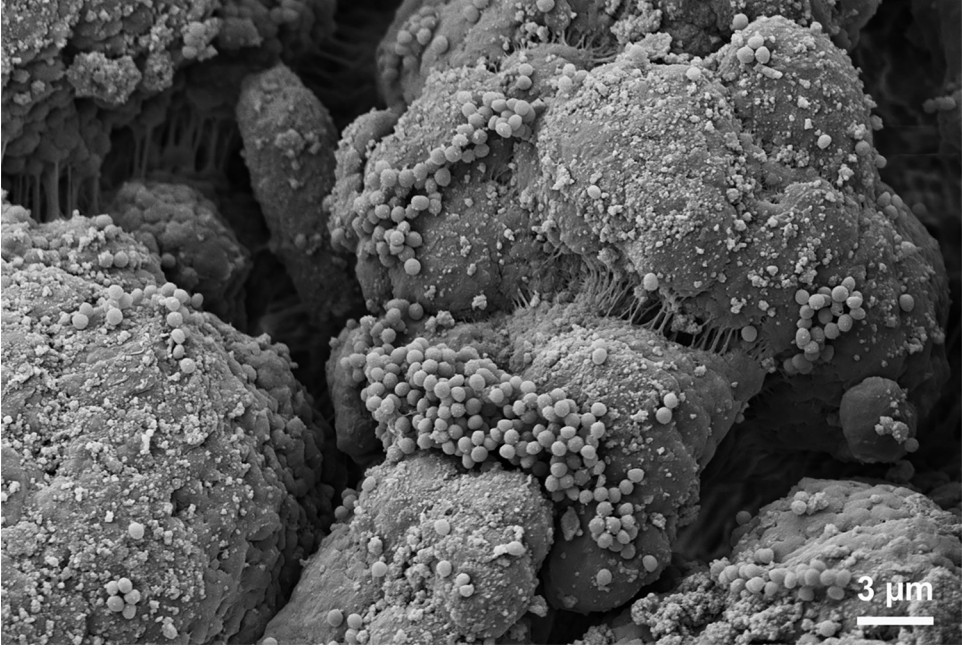

**Fig 3. Scanning electron microscopy of *S. aureus* Newman attached to heparin-coated beads.** Approximately $1.5 \times 10^8$ bacteria were added to 25 mg of the Seraph's material and fixed with 2% glutaraldehyde.

**Table 3. Binding activity of different *S. aureus* mutants to Seraph microcolumns.**

| | original bacterial solution | bound bacteria | | |
|---|---|---|---|---|
| | (total bacterial cell number ± SEM) | (total bacterial cell number ± SEM) | t-test (p-value) | percent removal |
| wild-type | $1.93 \times 10^6 \pm 6{,}51 \times 10^4$ | $1.17 \times 10^6 \pm 7.64 \times 10^4$ | + (0.0214) | 61% |
| *sigB* mutant | $1.81 \times 10^6 \pm 1{,}13 \times 10^5$ | $9.97 \times 10^5 \pm 1.57 \times 10^5$ | + (0.0186) | 55% |
| *saeS* mutant | $1.79 \times 10^6 \pm 7.73 \times 10^4$ | $1.12 \times 10^6 \pm 6.67 \times 10^4$ | + (0.0051) | 62% |
| *sarA* mutant | $9.56 \times 10^5 \pm 5.48 \times 10^4$ | $6.68 \times 10^5 \pm 3.49 \times 10^4$ | + (0.0012) | 70% |
| *lgt* mutant | $1.96 \times 10^6 \pm 6.80 \times 10^4$ | $1.06 \times 10^6 \pm 2.96 \times 10^4$ | + (0.0039) | 54% |

63%. In each case, t-test analyses showed that the treatment with Seraph had a statistically significant effect on the bacterial cell counts (p-values: Δ*lgt* = 0.0039, Δ*sigB* = 0.0186, Δ*sarA* = 0.0012, Δ*saeS* = 0.0051). The same was true for the wild-type, which showed a 61% reduction in cell number after treatment with Seraph (p-value = 0.0214) (**Tables 3 and S2**). A comparison of the percentage of bound bacteria of the wild-type and the different mutants using ANOVA showed no statistically significant differences (**S2 Table**).

## *S. aureus* binding activity to Seraph® 100 during a prolonged treatment period

To study *S. aureus* binding activity under conditions that mimic the clinical situation, we used the Seraph® 100 Microbind® Affinity blood adsorber connected to a hemofiltration device and a tubing system. The number of *gfp*-expressing *S. aureus* cells in the blood plasma was clearly reduced during treatment with the Seraph® 100 (**Fig 4**). A first sample taken 10 minutes after the start of the treatment showed that the number of bacteria in the plasma had decreased by approx. 33%. However, this value still varied between the different donors and the reduction did not reach statistical significance at this time (p-value = 0.22). After two hours of treatment, the percent removal of bacteria was approx. 64% and statistically significant. At this point, using a blood flow rate of 60 ml/min, each blood plasma sample had passed through the filter 7.2 times and $4.2 \times 10^8$ *gfp*-expressing *S. aureus* cells were bound to the filter (**S3 Table**). Two hours later, $5.5 \times 10^8$ bacteria were attached to the filter and the percent removal was 88% (**Fig 4**). No significant change in the number of *gfp*-expressing bacteria in the plasma was observed at later time points. For the present approach, this means that the maximum binding capacity of the Seraph® 100 is approx. $5.8 \times 10^8$ *S. aureus* cells, which corresponds to $5.8 \times 10^6$ *S. aureus* cells/ml beads. The resulting final mean percent removal was approx. 90% (**S3 Table**). This was achieved between four and six hours after the start of treatment (**Fig 4, S3 Table**).

## Discussion

The Seraph® 100 Microbind® Affinity Blood Filter (ExThera Medical Corp., Martinez, CA, USA) is the first CE-approved extracorporeal filter in use to eliminate bacteria from human blood. Ultra-high molecular weight polyethylene beads with endpoint-attached heparin are utilized to reduce the bacterial load in blood. The novel findings of this study are: i) components present in human blood plasma clearly interfere with the binding of *S. aureus* to heparin-coated beads, ii) differential expression of surface-associated proteins did not significantly

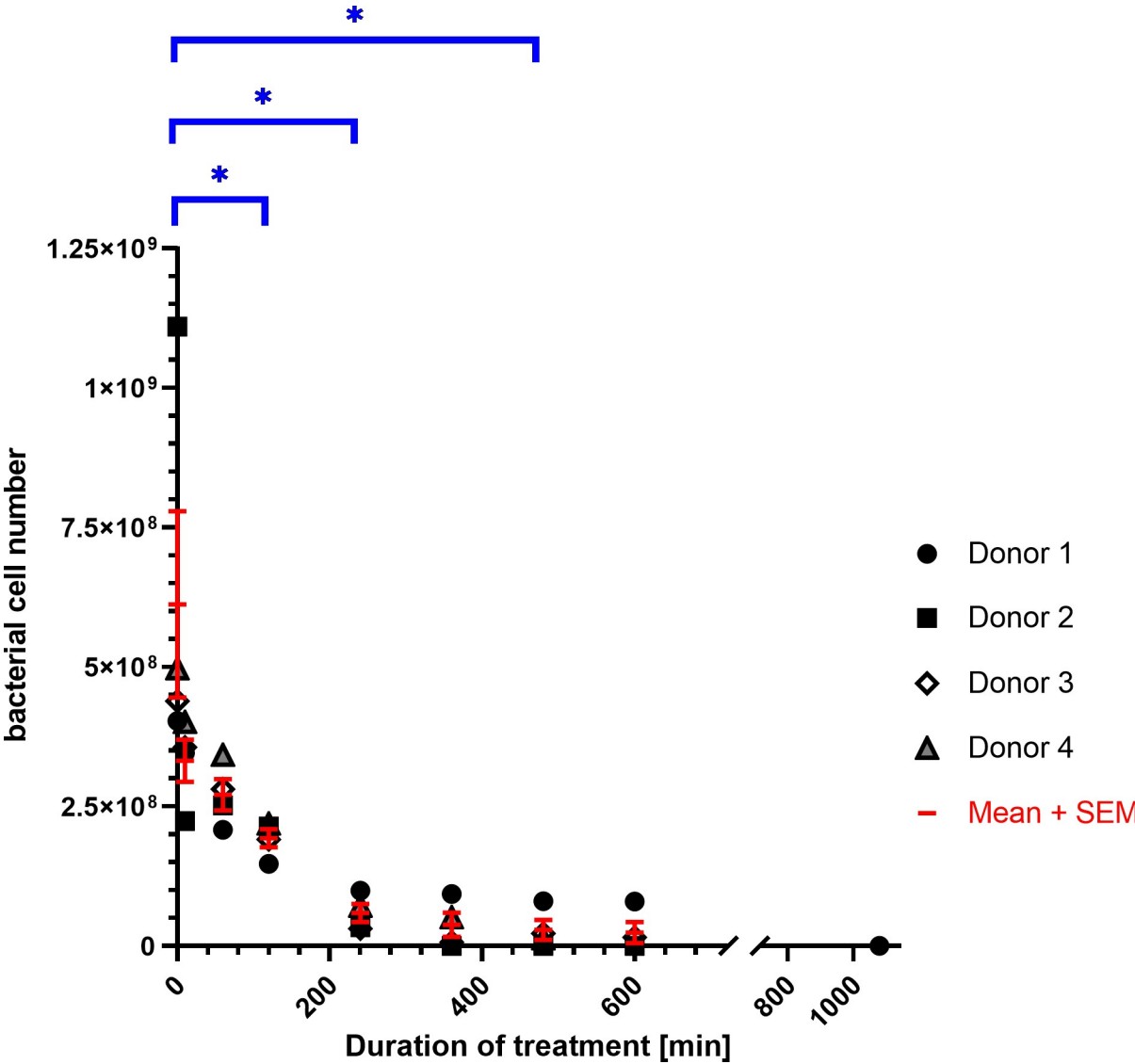

**Fig 4. *S. aureus* binding activity to Seraph® 100 during a prolonged treatment period.** Plasma samples of 1000 mL obtained from four different donors were inoculated with approx. 6.1 x 10⁸ bacteria and circulated through the Seraph® 100 Microbind® Affinity blood adsorber. Samples were taken before circulation (0 min) and at different time points after the start of circulation (10 min, 60 min. 120 min, 240 min, 360 min, 480 min, 600 min). The number of *gfp*-expressing bacteria was determined for each sample by FACS. Experiments were performed with human blood plasma from four different donors. The number of unbound bacteria for each donor and each sampling point is presented. In addition, the mean value of the unbound bacteria is shown. ANOVA and paired t-test were used to test whether the cell numbers changed significantly before and at different time points after the start of circulation. After 120, 240 and 480 minutes of circulation, the reduction in bacterial counts is statistically significant (p-values: 120 min = 0.01165, 240 min = 0.01344, 480 min = 0.04408) and marked with *.

affect the binding activity of *S. aureus* to heparin-coated beads and iii) final percentage removal of 90% for inoculated *S. aureus* in 1000 mL of human blood plasma was achieved between four and six hours of treatment with Seraph® 100 at a blood flow rate of 60 mL/min and showed a donor variance.

The reduction of *S. aureus* by Seraph micro columns was studied in 0.9% NaCl and in human blood plasma. Up to 60% of the bacteria were removed from inoculated NaCl. This is much less than in the case with *Escherichia coli*, where the bacterial cells were almost

completely removed after only one passage through the column [18]. The lower binding activity of *S. aureus* to the heparin-associated beads in blood plasma (percent removal = 10%) compared to NaCl suggests that there are proteins or other components in the plasma that somehow interfere with *S. aureus* binding to heparin. They may bind directly the heparan sulfate on the beads and thus compete with *S. aureus* for binding to these molecules, or they may adhere to the surface of *S. aureus* and sterically prevent the heparin-binding proteins expressed on the surface of *S. aureus* from accessing the heparan sulfate on the beads. In addition, the heparin added to the blood samples as an anticoagulant may also reduce the affinity of *S. aureus* for the heparin-coated beads by filling the heparin-binding sites on the bacterial surface.

*S. aureus* binding to heparin has been shown to be a multifactorial process depending on multiple surface associated proteins [43–45]. Using different mutants with global defects in the expression of surface associated proteins in *S. aureus*, the present study shows that the very different composition of surface proteins does not lead to drastic changes in the reduction rate of *S. aureus* by Seraph. These results suggest that heparin-coated beads are similarly effective at removing different clinical *S. aureus* isolates from patient blood, regardless of their surface-associated protein composition. This is also supported by a previous study showing that MRSA (n = 6) and MSSA (n = 1) isolates in whole blood bind to heparin with similar affinity [19].

In a more clinically relevant approach where the Seraph® 100 Microbind® Affinity blood adsorber was connected to a hemofiltration device and a tubing system, we studied the binding of *S. aureus* to the adsorber over time (= 10 hours) using plasma samples from four different patients and a bacterial flow rate of 60 mL/min. During this treatment, the plasma samples (= 1000 mL), spiked with approx. $6.1 \times 10^8$ *S. aureus* cells, passed through the adsorber 36 times. In the first phase, the number of bacteria in the plasma was continuously reduced. After approximately four hours, the maximum reduction in bacteria was almost achieved. Thereafter, the bacterial load in the plasma remained relatively constant and no significant release of bacteria into the plasma was observed, which could be caused, for example, by the growth of bacteria bound to the beads. This is an important observation, as it has previously been shown that bacteria on beads are still viable [19].

*S. aureus* and coagulase negative isolates incorporate heparin into the biofilm matrix, which significantly increases the biofilm capacity [43,46,47]. This is an undesirable and often problematic side effect and can also play a significant role in the success of treatment with Seraph. With our approach, we can clearly estimate that there is no increase in the bacterial load for a treatment period of up to ten hours. As *S. aureus* Newman is unable to form a robust biofilm [48], it is not possible to conclude from this study whether *S. aureus* biofilm formation has a significant effect on *S. aureus* reduction by Seraph. However, it is not expected that the *S. aureus* cells bound to the heparin-coated beads will be able to form a massive biofilm during a treatment period of four hours, as has been used in previous clinical situations with Seraph [49,50]. If longer treatment times are planned, biofilm formation by the bacteria bound on the beads should be excluded.

Extrapolating from our results, life size Seraph® 100 is capable of binding a minimum of $5 \times 10^8$ *S. aureus* cells in human blood plasma. Assuming a bacterial load in the blood of patients with bacteremia of up to $1 \times 10^4$ CFUs/ml [51], which corresponds to a total bacterial count of $6 \times 10^5$ CFUs in an adult (6 L of blood), the filtering capacity of Seraph® for *S. aureus* should be sufficient to reduce the number of *S. aureus* to a level that can be processed by the immune system. At a total flow rate of 60 mL/min, the total volume of blood of an adult patient would have passed through the filter once after 100 min of use. Our *in vitro* data showed that at least $2.6 \times 10^8$ bacteria were bound to the filter after 120 min of treatment representing a 64%

reduction in bacterial load. It is therefore tempting to speculate that at least half of the bacteria present in the blood of patients with bacteraemia should be attached to the filter after the first passage.

Based on our results, extracorporeal blood purification using heparin-coated beads is a highly reliable and efficient treatment strategy for *S. aureus* bloodstream infections and represents a promising alternative for infections with multi-resistant strains. Treatment durations of up to four hours may be sufficient to reduce the number of bacterial cells to a level that the immune system can handle. In the first in-patient applications, patients with persistent blood stream infections were treated with the Seraph® 100 Microbind® Affinity Blood Filters for four hours at a blood flow rate of 225 to 300 ml/min, which is significantly higher than in the present study. A reduction in bacterial counts was observed, but this varied widely between patients and did not reach statistical significance [49,50]. There are many reasons for the high variability between patients and the resulting differences in treatment outcomes. Our evidence suggests that both the composition of the blood plasma and the duration of the treatment affect the outcome. Other factors may include blood cell composition, bacterial load, and amount of heparin administered. Blood flow rate, temperature, blood pressure, the type of bacterial pathogen and the timing of treatment may also affect bacterial reduction in the clinical use of Seraph [50]. A systematic analysis of the influence of these parameters on bacterial reduction during haemodialysis is still lacking, but is essential to better assess the effectiveness of the treatment in the future.

## Supporting information

**S1 Table. Binding of *S. aureus* Newman to Seraph micro columns.**
(XLSX)

**S2 Table. Binding of different *S. aureus* mutants to Seraph micro columns.**
(XLSX)

**S3 Table. *S. aureus* binding activity to Seraph during a prolonged treatment period.**
(XLSX)

## Acknowledgments

We thank ExThera Medical Corporation for supplying Seraph adsorber mini columns and Seraph® 100 Microbind® Affinity blood adsorber. We acknowledge Julius Schmidt (Hannover Medical School) for providing plasma samples and Martin Kucklick for support in statistical analysis. We are grateful to Richard A. Proctor for reviewing the manuscript.

## Author Contributions

**Conceptualization:** Jan T. Kielstein, Susanne Engelmann.

**Data curation:** Malin-Theres Seffer, Martin Weinert.

**Formal analysis:** Malin-Theres Seffer, Martin Weinert.

**Investigation:** Malin-Theres Seffer, Martin Weinert, Gabriella Molinari.

**Methodology:** Martin Weinert, Gabriella Molinari, Manfred Rohde, Lothar Gröbe.

**Project administration:** Jan T. Kielstein, Susanne Engelmann.

**Resources:** Manfred Rohde, Lothar Gröbe, Jan T. Kielstein, Susanne Engelmann.

**Supervision:** Jan T. Kielstein, Susanne Engelmann.

**Validation:** Malin-Theres Seffer.

**Visualization:** Malin-Theres Seffer, Gabriella Molinari, Manfred Rohde.

**Writing – original draft:** Malin-Theres Seffer, Susanne Engelmann.

**Writing – review & editing:** Malin-Theres Seffer, Martin Weinert, Gabriella Molinari, Manfred Rohde, Lothar Gröbe, Jan T. Kielstein.

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
