## [Decision Letter · Decision Letter 0]

20 Jan 2023

PONE-D-22-34334Staphylococcus aureus binding to Seraph® 100 Microbind® Affinity Filter: Effects of surface protein expression and treatment durationPLOS ONE

Dear Dr. Engelmann,

Thank you for submitting your manuscript to PLOS ONE. After careful consideration, we feel that it has merit but does not fully meet PLOS ONE’s publication criteria as it currently stands. Therefore, we invite you to submit a revised version of the manuscript that addresses the points raised during the review process.

Please submit your revised manuscript by Mar 06 2023 11:59PM. If you will need more time than this to complete your revisions, please reply to this message or contact the journal office at plosone@plos.org. Please include the following items when submitting your revised manuscript:A rebuttal letter that responds to each point raised by the academic editor and reviewer(s). You should upload this letter as a separate file labeled 'Response to Reviewers'.A marked-up copy of your manuscript that highlights changes made to the original version. You should upload this as a separate file labeled 'Revised Manuscript with Track Changes'.An unmarked version of your revised paper without tracked changes. You should upload this as a separate file labeled 'Manuscript'.If applicable, we recommend that you deposit your laboratory protocols in protocols.io to enhance the reproducibility of your results. Protocols.io assigns your protocol its own identifier (DOI) so that it can be cited independently in the future. For instructions see: https://journals.plos.org/plosone/s/submission-guidelines#loc-laboratory-protocols. Additionally, PLOS ONE offers an option for publishing peer-reviewed Lab Protocol articles, which describe protocols hosted on protocols.io. Read more information on sharing protocols at https://plos.org/protocols?utm_medium=editorial-email&utm_source=authorletters&utm_campaign=protocols.

We look forward to receiving your revised manuscript.

Kind regards,

Awatif Abid Al-Judaibi, PhD

Academic Editor

PLOS ONE

Journal Requirements:

Reviewers' comments:

Reviewer's Responses to Questions

**Comments to the Author**

1. Is the manuscript technically sound, and do the data support the conclusions?

Reviewer #1: Yes

Reviewer #2: Yes

2. Has the statistical analysis been performed appropriately and rigorously? 

Reviewer #1: Yes

Reviewer #2: I Don't Know

3. Have the authors made all data underlying the findings in their manuscript fully available?

Reviewer #1: Yes

Reviewer #2: Yes

4. Is the manuscript presented in an intelligible fashion and written in standard English?

Reviewer #1: No

Reviewer #2: Yes

5. Review Comments to the Author

Reviewer #1: PONE-D-22-34334

Abstract section

1. Please include the level of significant in the abstract section. i.e The result obtained shows a significant effect of ……………………….. (p ≤ 0.05).

2. I therefore suggest that the abstract should be rewritten in the following pattern to illuminate on key finding from this study.

a. Brief introduction

b. Aim and objectives

c. Key and most important results

d. Conclusion and contribution to knowledge.

e. Please provide a more quantitative data rather more descriptive data.

Introduction

1. Please kindly recast this sentence ‘Sepsis is a serious worldwide health threat and affects annually 30 to 50 million people worldwide’’. You can change it to Sepsis is a serious worldwide health threat that has been reported to affects annually 30 to 50 million people worldwide.

2. Please kindly recast this ‘’Hence, the WHO ranks sepsis a “key issue for global health”

3. Please kindly recast this and don’t start a sentence with ‘’Increasing’’ from this sentence ‘’Increasing antibiotic resistance in bacteria further complicates treatment of these infections.’’

4. Please kindly recast this ‘’Staphylococcus aureus is one of the most frequent and dangerous pathogens that, in addition to mild skin and soft tissue infections, can cause systemic infections as pneumonia, endocarditis or sepsis’’

5. Please kindly recast this ‘’Patients after surgery or those receiving continuous ambulatory peritoneal dialysis or hemodialysis have a higher risk to get infected with this pathogen’’

6. The authors needs to clearly states the aims and objective of this study very clearly. You need to start this in a new paragraph.

Result

6. Please kindly provide a recent reference to support the following experiment by saying this experiment was carried out using the methodology developed by (Ten et al., 2023)

a.S. aureus binding experiments with miniaturized Seraph adsorbers (micro columns)

b.S. aureus binding experiments with Seraph® 100 Microbind® Affinity blood adsorber

c.Bacteria cell counting and enumeration assay

7. I would like to encourage the author's to clarify the experimental design. A clear experimental design with the treatments and replicates clearly described is mandatory for the acceptance of the manuscript. The statistical analysis and corresponding results also need to be clearly presented. Please add the F values, the degrees of freedom and the precise p values obtained. The assumptions of ANOVA were tested? The software was manufactured in which year?

8. The author should try to better show the results of the statistical analysis, at least indicating the higher significant p-value obtained for each test (e.g. p ≤ 0.0…) throughout the result section.

9. Please improve the quality and the resolution of figures. Some of them are very hard to read/see-.Most especially figure 2,34.

10. Please include the level of stastical analysis at the figure caption of each figures most especially figure

11. Please provide a more quantitative data rather more descriptive data.

12. Please include the level of significant in figure and also Please include the level of stastical analysis at the figure caption of each figures most especially the figures

13. Please replace the old references

14. Please provide the geographical location where the experiment was carried out i.e town (8°8′0″N, 4°16′0″E).

Discussion section

1. The author need to relate the results obtained during this study with relevant discussion and compare the results obtained to previously results from other researchers

2. The manuscript is based on a very good concept methodologically executed but poorly written. The methods failed to align with the result with the discussion. Authors need to surrender this paper to serious editorial review by an English expert or language skilled colleague. This would illuminate the manuscript and makes it more comprehensive.

References

The authors should check the reference if they are in accordance with the format stipulated by the journal.

Reviewer #2: The study describes the binding affinity of Staphylococcus aureus to the Seraph® 100 Microbind® Affinity Filter, and analyzed the effects of surface protein expression and treatment duration on the rate of binding and population reduction in the original substrate.

The manuscript was generally well written, and the methods used were appropriate.

The introduction section was well written and captures the background of the study appropriately.

Again the materials and methods were explained in detailed and allows for reproducibility.

However, the result section was not explicit enough as a lot of the methods were repeated, and a significant amount of the data generated were left out, and not comprehensively expressed in the result section.

Also, inferences that should be explained and discussed in the discussion section were expressed under results.

The discussion also could be more explicit to drive home the points and findings of the authors.

The figures and tables were adequate and explanatory.

6. PLOS authors have the option to publish the peer review history of their article (what does this mean?). If published, this will include your full peer review and any attached files.

Reviewer #1: **Yes: **Prof Charles Oluwaseun Adetunji

Reviewer #2: No

---

## [Author Response · Author response to Decision Letter 0]

2 Mar 2023

Dear Editor,

Dear Reviewers,

We are very grateful for your comments and suggestions, which have been very helpful in significantly improving our manuscript. We respond to each comment separately below. 

Editor:

Answer: The PLOS ONE's style requirements have been applied to the entire manuscript, to the tables and to the figures.

Answer: A new section for supporting information files has been added at the end of the manuscript and all supporting files are now described there.

Answer: We have carefully checked all references in the results section and there was no retracted paper cited. We have added three new references to the reference list. 

Reference No 3: This reference was only cited in the text in the earlier version of the manuscript. It is a WHO report and is only published on the WHO website. We have now added it to the reference list as an online article.

Reference No 25: This reference is new to the manuscript. We have included the reference in the manuscript to describe more precisely, which S. aureus wild-type strains we used for our experiments. 

Reference No. 50: This reference is also new to the manuscript and was not published until the end of 2022. This is the first major clinical study on the use of Seraph in haemodialysis patients with blood stream infections and highly relevant to our study. In the discussion section of the manuscript, we discuss our findings in relation to the data from this clinical trial.

Reviewer #1:

Abstract section

1. Please include the level of significant in the abstract section. i.e The result obtained shows a significant effect of ……………………….. (p ≤ 0.05).

Response: The level of significance of the data and p-values were inserted in the abstract section at appropriate places.

2. I therefore suggest that the abstract should be rewritten in the following pattern to illuminate on key finding from this study.

a. Brief introduction

b. Aim and objectives

c. Key and most important results

d. Conclusion and contribution to knowledge.

e. Please provide a more quantitative data rather more descriptive data.

Response: The Abstract section has been completely revised according to the reviewer´s recommendations. The suggested structure has been adopted and quantitative data are now presented together with statistical information.

Introduction

1. Please kindly recast this sentence ‘Sepsis is a serious worldwide health threat and affects annually 30 to 50 million people worldwide’’. You can change it to Sepsis is a serious worldwide health threat that has been reported to affects annually 30 to 50 million people worldwide.

Resonse: The sentence has been changed accordingly (lines 49-52).

2. Please kindly recast this ‘’Hence, the WHO ranks sepsis a “key issue for global health”

Response: The sentence has been rearranged (lines 52-53).

3. Please kindly recast this and don’t start a sentence with ‘’Increasing’’ from this entence ‘’Increasing antibiotic resistance in bacteria further complicates treatment of these infections.’’

Response: The sentence has been rearranged (lines 54-56).

4. Please kindly recast this ‘’Staphylococcus aureus is one of the most frequent and dangerous pathogens that, in addition to mild skin and soft tissue infections, can cause systemic infections as pneumonia, endocarditis or sepsis’’

Response: The sentence has been rearranged (lines 57-59).

5. Please kindly recast this ‘’Patients after surgery or those receiving continuous ambulatory peritoneal dialysis or hemodialysis have a higher risk to get infected with this pathogen’’

Response: The sentence has been rearranged (lines 59-61).

6. The authors needs to clearly states the aims and objective of this study very clearly. You need to start this in a new paragraph.

Response: As suggested by the reviewer, we have now formulated the objectives somewhat more clearly in a separate paragraph (lines 84-90).

Result

6. Please kindly provide a recent reference to support the following experiment by saying this experiment was carried out using the methodology developed by (Ten et al., 2023)

a.S. aureus binding experiments with miniaturized Seraph adsorbers (micro columns)

b.S. aureus binding experiments with Seraph® 100 Microbind® Affinity blood adsorber

c.Bacteria cell counting and enumeration assay

Response: The existing references have been referred to even more clearly, and any deviations from the procedure described there have been briefly presented. ((b) lines 136-142, (c) lines 157-165)

However, there is no reference for the experiment “S. aureus binding with miniaturized Seraph adsorbers” (a). 

7. I would like to encourage the author's to clarify the experimental design. A clear experimental design with the treatments and replicates clearly described is mandatory for the acceptance of the manuscript. The statistical analysis and corresponding results also need to be clearly presented. Please add the F values, the degrees of freedom and the precise p values obtained. The assumptions of ANOVA were tested? The software was manufactured in which year?

Response: The Material and Methods section has been revised according to the reviewer´s recommendations. 

Statistical analysis are now described more clearly and the used software version is indicated (lines 184-192)

8. The author should try to better show the results of the statistical analysis, at least indicating the higher significant p-value obtained for each test (e.g. p ≤ 0.0…) throughout the result section.

Response: In Tables 2 (line 227) and 3 (line 249), which show the data from the “micro column” experiments, we have now also added the p-values and the SEM values. In addition, the data used for the statistics and the results of the statistics are presented in even more detail in S1 and S2 Tables, which are new to the manuscript. S3 Table (formerly S1 Table) shows all relevant data and the results of the statistics for the “prolonged treatment period” experiment. Means and SEMs for the “prolonged treatment period experiment” are now also shown in Figure 4 and all relevant information for the statistics is given in the figure legend (lines 270-281). 

9. Please improve the quality and the resolution of figures. Some of them are very hard to read/see-.Most especially figure 2,34.

Response: The figures were revised according to the PLOS One requirements and the resolution of Figure 3 was increased.

10. Please include the level of stastical analysis at the figure caption of each figures most especially figure

Response: This applies to Figure 4. All information relevant to the statistics is now given in the figure legend (lines 270-281). 

11. Please provide a more quantitative data rather more descriptive data.

Response: As recommended, much more quantitative data is now presented together with statistical information in the Results section and in Table 4 and in S1, S2 and S3 Tables.

12. Please include the level of significant in figure and also Please include the level of stastical analysis at the figure caption of each figures most especially the figures

Response: This applies to Figure 4. All information relevant to the statistics is now given in the figure legend (lines 270-281). 

13. Please replace the old references

Response: We have carefully checked all references in the results section and we believe we have included the most relevant articles.

14. Please provide the geographical location where the experiment was carried out i.e town (8°8′0″N, 4°16′0″E).

Response: This is not relevant to this study. Experiments were performed under standard laboratory conditions.

Discussion section

1. The author need to relate the results obtained during this study with relevant discussion and compare the results obtained to previously results from other researchers

Response: The Discussion section has been rigorously revised and rewritten as recommended by the reviewer. We have tried to make the reference to previously published data even clearer. We have also included an additional reference in the discussion of the data, which was not published until the end of 2022 (Eden et al 2022, Reference [50]). This is the first major clinical study on the use of Seraph in haemodialysis patients with blood stream infections.

2. The manuscript is based on a very good concept methodologically executed but poorly written. The methods failed to align with the result with the discussion. Authors need to surrender this paper to serious editorial review by an English expert or language skilled colleague. This would illuminate the manuscript and makes it more comprehensive.

Response: The manuscript has been proofread by a native speaker. We hope that it is now a little clearer and a little easier to understand. 

References

The authors should check the reference if they are in accordance with the format stipulated by the journal.

Response: The references have been checked and are now in accordance with the PLOS One reference style requirements.

Reviewer #2: 

The study describes the binding affinity of Staphylococcus aureus to the Seraph® 100 Microbind® Affinity Filter, and analyzed the effects of surface protein expression and treatment duration on the rate of binding and population reduction in the original substrate.

The manuscript was generally well written, and the methods used were appropriate.

The introduction section was well written and captures the background of the study appropriately.

Again the materials and methods were explained in detailed and allows for reproducibility.

Response: Thank you very much for this nice evaluation.

However, the result section was not explicit enough as a lot of the methods were repeated, and a significant amount of the data generated were left out, and not comprehensively expressed in the result section.

Answer: We have largely removed the experimental aspects from the Results section and present them now exclusively in the Methods sections.

Also, inferences that should be explained and discussed in the discussion section were expressed under results.

Answer: We have now made a clearer distinction between the Results and the Discussion sections, in particular moving conclusions from the Results to the Discussion section. 

The discussion also could be more explicit to drive home the points and findings of the authors.

Answer: We have restructured the discussion section somewhat and revised it considerably. We hope that it is now a little clearer.

The figures and tables were adequate and explanatory.

---

## [Editor Report · Decision Letter 1]

6 Mar 2023

Staphylococcus aureus binding to Seraph® 100 Microbind® Affinity Filter: Effects of surface protein expression and treatment duration

PONE-D-22-34334R1

Dear Dr. Susanne Engelmann,

We’re pleased to inform you that your manuscript has been judged scientifically suitable for publication and will be formally accepted for publication once it meets all outstanding technical requirements.

Kind regards,

Awatif Abid Al-Judaibi, PhD

Academic Editor

PLOS ONE

---

## [Editor Report · Acceptance letter]

9 Mar 2023

PONE-D-22-34334R1 

*Staphylococcus aureus* binding to Seraph® 100 Microbind® Affinity Filter: Effects of surface protein expression and treatment duration 

Dear Dr. Engelmann:

I'm pleased to inform you that your manuscript has been deemed suitable for publication in PLOS ONE. Congratulations! Your manuscript is now with our production department. 

Kind regards, 

on behalf of

Professor Awatif Abid Al-Judaibi 

Academic Editor

PLOS ONE